# A Multi-Task Classification Method for Application Traffic Classification Using Task Relationships

**Ui-Jun Baek** [1], **Boseon Kim** [2], **Jee-Tae Park** [1], **Jeong-Woo Choi** [1] and **Myung-Sup Kim** [1,*]

[1] Department of Computer and Information Science, Korea University, Sejong-si 30019, Republic of Korea; pb1069@korea.ac.kr (U.-J.B.); pjj5846@korea.ac.kr (J.-T.P.); choigoya97@korea.ac.kr (J.-W.C.)
[2] Korea Institute of Science and Technology Information, Daejeon 34141, Republic of Korea; boseon12@kisti.re.kr
\* Correspondence: tmskim@korea.ac.kr

**Abstract:** As IT technology advances, the number and types of applications, such as SNS, content, and shopping, have increased across various fields, leading to the emergence of complex and diverse application traffic. As a result, the demand for effective network operation, management, and analysis has increased. In particular, service or application traffic classification research is an important area of study in network management. Web services are composed of a combination of multiple applications, and one or more application traffic can be mixed within service traffic. However, most existing research only classifies application traffic by service unit, resulting in high misclassification rates and making detailed management impossible. To address this issue, this paper proposes three multitask learning methods for application traffic classification using the relationships among tasks composed of browsers, protocols, services, and application units. The proposed methods aim to improve classification performance under the assumption that there are relationships between tasks. Experimental results demonstrate that by utilizing relationships between various tasks, the proposed method can classify applications with 4.4%p higher accuracy. Furthermore, the proposed methods can provide network administrators with information about multiple perspectives with high confidence, and the generalized multitask methods are freely portable to other backbone networks.

**Keywords:** application traffic classification; network management; multitask learning



## 1. Introduction

With the recent advancement of IT technology, web services have become increasingly important in daily life, and due to the influence of COVID-19, the use of video streaming and online shopping has dramatically increased as indoor activity time has prolonged [1]. As a result, the demand for monitoring and analyzing network traffic, including application traffic classification and traffic prediction, has increased due to the emergence of complex and diverse application traffic resulting from the increase in the number and types of applications, such as SNS (Social Network Service), content streaming, and shopping. In particular, application traffic classification research is essential for effective network monitoring and analysis [2]. It can be widely used in areas such as cloud service pricing, resource planning, traffic control, and network security. For instance, in schools or public institutions, network resources can be restricted to limit non-work-related traffic, and companies subscribing to cloud services can classify the traffic of the services they use to subscribe to the appropriate services without unnecessary consumption.

Web services are software systems for application interaction between different types of computers on the network and can be composed of a combination of multiple applications. Therefore, the traffic generated by web services is also composed of a combination of traffic generated by various applications. However, most existing research only classifies network traffic by service unit and application unit, and this approach is similar to MCC

(Multiclass Classification) shown in Figure 1. This can result in misclassification of mixed traffic in a service that includes multiple services or application traffic. For example, the traffic flow of *Googlefonts* in the *Naver* service and the traffic flow of *Googlefonts* in the *YouTube* service represent two different ground truths, even though they are both under the same sub-service. This can confuse the learning model when learning the characteristics of the traffic. Moreover, simply classifying multiple sub-service traffic or application traffic within a service only by service unit makes detailed management impossible. To address this issue, this paper proposes a method for classifying traffic using the relationships among four tasks, as shown in MTC (Multitask Classification) in Figure 1. Multitask learning (MTL) [3] is applied in a variety of fields, with the aim of simultaneously learning multiple related tasks so that the knowledge contained in one task is used for other tasks to improve the generalization performance of all tasks [4]. Ref. [5] performs multitask learning through the task lists provided by CICIDS 2017 [6], ISCX VPN-nonVPN 2016 [7], and ISCX Tor 2016 [8]. The task lists include normal/abnormal application categories, detailed applications, encryption, etc. Ref. [9] performed multitask learning through the task list provided by ISCX VPN-nonVPN 2016, which includes a total of three task lists. Ref. [10] performed multitask learning by creating a new task called Bandwidth and Duration from the QUIC and ISCX datasets. Unfortunately, there are not many multitask-based classification methods in the field of network traffic classification, and this is also often dependent on the task list provided by the dataset. Therefore, various tasks that can improve generalization performance in the field of network traffic classification need to be proposed. We set the goal of improving classification performance under the assumption that the tasks are not completely independent and have relationships among them. For instance, when using the *Edge* browser, a web browser released by *Microsoft*, *Microsoft* traffic mainly occurs when using the *Edge* browser to access web services. Similarly, when using the *Firefox* browser, *Mozilla* traffic mainly occurs when using the *Firefox* browser to access web services. In another case, the *YouTube* service communicates using the HTTP/3 protocol, and most of the traffic using the HTTP/3 protocol occurs within the *YouTube* service. The proposed method includes four tasks for traffic classification, and classifies accurate services and sub-services or applications by performing four tasks simultaneously using MTC. In addition, the proposed method provides detailed classification results for traffic, which can satisfy various requirements of network administrators. Our representative contributions include the following:

(i). Improved classification accuracy: improved classification performance considering relationships between multiple tasks (browsers, HTTP protocols, applications, services);

(ii). Generalizability and portability of the four multitask classification methods: the generalized classification model for multiple classifications improves classification performance across diverse backbone networks;

(iii). Possibility to monitor and analyze from multiple perspectives: network administrators can gain more detailed information and insights into the traffic occurring on the networks under their jurisdiction when monitoring and analyzing their networks.

This paper is structured as follows: Related research is described in Section 2 following the introduction. Section 3 provides a detailed explanation of the proposed method, and Sections 4 and 5 describe the experimental setup and results, respectively. Finally, in Section 6, the conclusion and future research are discussed, concluding this paper.

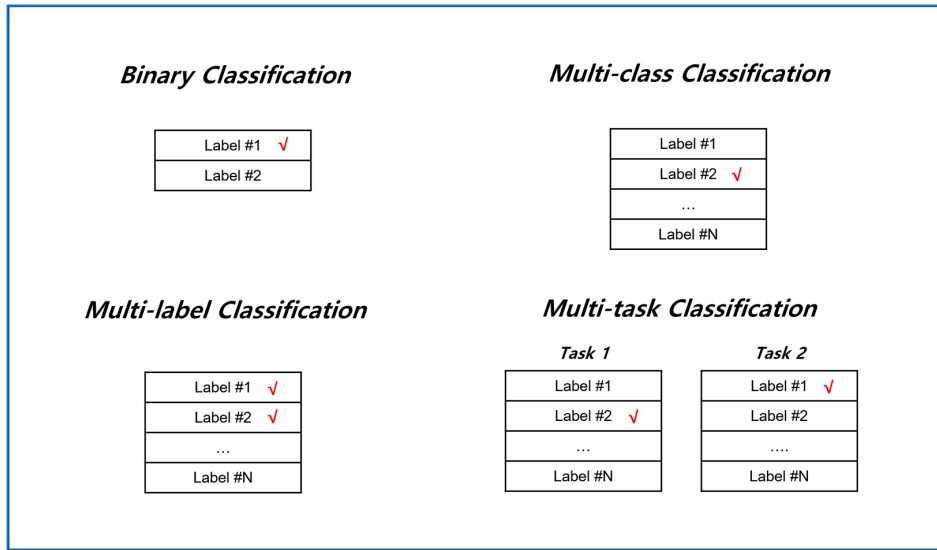

**Figure 1.** Overview of classification types.

## 2. Related Works

### 2.1. Task Description

This subsection describes four tasks of the proposed methodology. The first task is browser classification, where browsers are graphical user interface-based software applications that enable bidirectional communication between users and web servers, allowing the display of HTML documents or files. The labels included in the browser classification task are *Chrome*, *Edge*, and *Firefox*, whose combined usage accounts for the majority of global browser usage. Ref. [11] proposed real-time lightweight identification of HTTPS clients based on network monitoring and SSL/TLS fingerprinting and reported that 95.4% of HTTPS network traffic could be retrieved by the proposed method. Additionally, the study reported that the handshake fingerprints of SSL/TLS, including the cipher suite list of different clients, differ. This indicates that prior information about the browser or communication client can influence the classification results. The second task is protocol classification, where collected protocols include HTTP/1.1, HTTP/2 (HTTPS), and HTTP/3. HTTP/1.1 is one of the HTTP protocol versions released in 1999 and is still the most widely used. HTTPS refers to the second major version of the HTTP protocol, which was released in 2015. HTTP/3 is the third major version of the HTTP protocol, which was released in 2020, and uses the QUIC (Quick UDP Internet Connections) protocol instead of the TCP protocol used in previous versions, providing faster and more reliable data transfer. Web applications utilize various HTTP protocol versions, and the protocol version can serve as useful prior information for classifying specific applications. Ref. [12] proposes a method to improve the service classification performance by using the protocol classification results of the application traffic test dataset as prior information in the service classification process. The third task is service classification, where services are defined as software systems for interaction between different types of computers on the network, consisting of *Aladin*, *Amazon*, *Google*, *Nate*, *Naver*, and *YouTube*. For convenience, the services referred to in this paper denote the top-level service that includes multiple applications or sub-services. The fourth task is application classification, where applications or sub-services responsible for specific interactions within a service are identified. For example, *Google Fonts* and *Gstatic* are applications (sub-services) provided by *Google*, while *search.naver* and *pstatic* are applications provided by *Naver*. In other words, a service may consist of multiple applications or sub-services. The four tasks selected in this study are closely interconnected, and when performing each task, the other tasks can serve as valuable prior information.

## 2.2. Classification Type

This subsection describes the types and definitions of classification tasks. Classification tasks can be divided into BC (Binary Classification), MCC (Multiclass Classification), MLC (Multi-label Classification), and MTC (Multitask Classification), as shown in Figure 1.

BC is a classification task with two classes, where each sample can be labeled with only one class. For example, in an anomaly classification task to distinguish between anomaly and benign, the user can assign a label to each sample with only one of the two classes. There have been many studies on detecting the presence of malicious traffic in network traffic data [13–15].

MCC is a classification task with more than two classes, where each sample can be labeled with only one class. The majority of research in the field of application traffic classification is focused on MCC, where various application traffic types are labeled with a single label. For example, in the application classification task [7], to distinguish between Mail, File Transfer, P2P, VoIP, Streaming, and Chat, the user can assign a label to each sample with only one of the six classes.

MLC is a classification task in which multiple labels are assigned to each sample, equal to the number of possible classes when there are multiple classes. For example, in a weather classification task that includes seven classes, such as clear, cloudy, snow, rain, fog, thunder, and hail, the user can assign one or more labels from the seven classes to each sample.

MTC is a Multiclass–Multioutput Classification. MTC is used in the proposed methodology, where there are multiple tasks, and the user can assign only one label for each task.

## 2.3. Structured Inference Neural Network

The Structured Inference Neural Network (SINN) was inspired by a deep learning-based method that utilizes various label relationships to improve image classification performance by using a cumulative label prediction neural network [16]. In this neural network, structural graph formation is possible through relationships between labels, and different interpretations of various units are possible for representing images. For example, an image can be represented in terms of indoor or outdoor, specific location, and specific object units. As a result, SINN is a structural inference neural network that can model relationships between labels by considering dependencies between classification units through CNN and RNN. Figure 2a shows a baseball field image that can be represented as a structural graph, as shown in Figure 2b. The baseball field in Figure 2a belongs to the scene unit's artificial outdoor, the scene attribute unit's sports field or artificial element, the detailed scene unit's home plate, and the object unit's field, baseball bat, baseball, grass, and person classes, which are all represented by the structural graph in Figure 2b and are represented by red nodes. If the image belongs to the indoor class at the scene unit, the baseball bat object cannot be present, as the baseball bat object is dependent on artificial outdoor, which serves as evidence for using SINN. Inspired by these structured representations, we propose four units (browser, protocol, application, service) to represent traffic flows and use them to perform MTC.

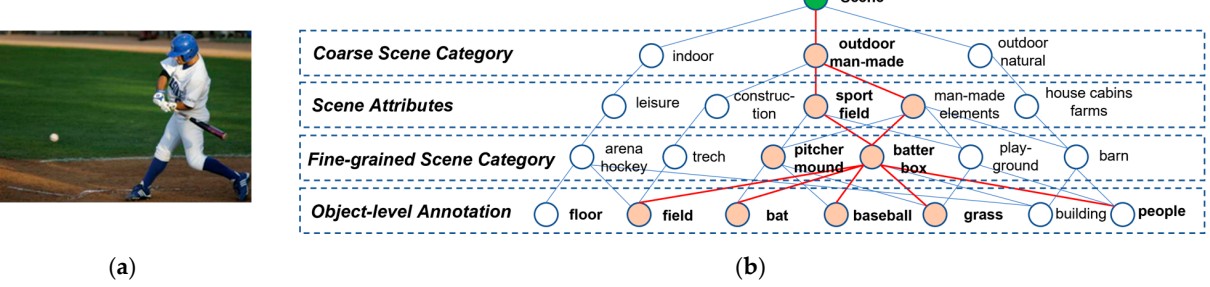

(**a**)                                    (**b**)

**Figure 2.** These represent a baseball field image (**a**) and its corresponding structural graph (**b**). (**a**) Baseball field image. (**b**) An image of a baseball field represented as a structural graph.

### 2.4. DL-Based Spatial-Temporal Feature Extraction

In this section, we describe the deep learning-based spatial and temporal feature extraction methods used in the multitask learning approach.

CNN is the most widely used method for extracting spatial features, especially for processing images or video data. CNNs are inspired by the structure of the visual cortex in animals, which has layers of neurons that are sensitive to specific visual features. Similarly, CNNs have layers that extract hierarchical representations of the input image or video, starting from low-level features, such as edges, and gradually moving towards high-level features, such as object parts and whole objects. The basic component of a CNN is the convolutional layer, which applies a set of filters or kernels to the input image, extracting local features that are then pooled and passed on to the next layer. We utilized two famous CNN-based backbone networks and two deep learning architectures proposed in previous studies to clearly demonstrate the contribution of applying the MTC method. The first one is Lenet [17], an initial model designed for handwritten digit recognition. Lenet consists of two convolutional layers, pooling layers and fully connected layers, and its model structure is shown in Figure 2. The second backbone network is Resnet [18] (Residual Network), a deep learning architecture proposed to solve the gradient vanishing problem that occurs in deep neural networks. Resnet solves the gradient vanishing problem by introducing skip connections, direct connections that skip several layers in the network, unlike traditional CNN architectures. Resnet still shows good performance in various fields. Ref. [19] proposes MISCNN (Multi-Input Shape Convolutional Neural Network) that utilizes various input forms that can be derived from fixed-length packet bytes. By observing packets from various angles through the different forms that can be derived from a single input, it shows a significant improvement in performance compared to previous research. Ref. [20] proposes HAST-IDS (Hierarchical Spatialtemporal Features-based Intrusion Detection System), an intrusion detection system that uses CNN to learn spatial features of packets and LSTM to learn temporal features between multiple packets. HAST-IDS performs a classification of multiple normal and abnormal traffic, and experiments show that HAST-IDS outperforms other approaches in terms of accuracy, detection rate, and FAR.

RNN (Recurrent Neural Network) is a type of deep learning that is used to handle sequential data [21]. RNN has the advantage of being able to solve the long-term dependency problem by using the output of the previous step as the input of the current step, thereby reflecting the previous information in the current processing. However, RNNs can suffer from vanishing gradient and exploding gradient problems. To solve this problem, a model based on RNN called GRU (Gated Recurrent Unit) was proposed [22], and this paper applies GRU to extract temporal features. GRU has the advantage of faster learning speed and the ability to handle longer sequences than RNN. GRU combines the hidden state and cell state used in RNN into one and updates it using two gates: the update gate and the reset gate. The update gate determines how much information to update using the current input and previous state, and the reset gate determines how much the previous state is forgotten. Through this, GRU can solve the long-term dependency problem while mitigating problems that arise during the learning process.

### 2.5. MTC-Based Traffic Classification

Ref. [5] proposes the use of multitask deep neural network in federated learning (MT-DNN–FL) to simultaneously perform network anomaly detection, VPN (Tor) traffic recognition, and traffic classification tasks. They report that the multitasking approach reduces training time overhead compared to multiple single-task models. Experimental results conducted on well-known datasets, such as CICIDS2017, ISCXVPN2016, and ISX-Tor2016, demonstrate that the proposed method achieves superior anomaly detection and classification performance compared to baseline models in a centralized training architecture. Ref. [10] proposes the use of multitask learning to predict the bandwidth and duration of network traffic flows while simultaneously classifying the traffic into different classes. Predicting bandwidth and duration does not require extensive labeling efforts or

specific environments, allowing for the utilization of abundant training data. This approach significantly reduces the number of labeled samples required for traffic class prediction. Furthermore, the predicted bandwidth and duration can be applied in ISPs for resource allocation, routing, and QoS purposes. The experiments conducted on the QUIC and ISCX VPN-nonVPN datasets demonstrate that the multitask learning approach outperforms single-task learning and transfer learning methods. Ref. [9] proposed a novel multimodal multitask deep learning approach called DISTILLER. This approach is designed to address the challenges of encrypted traffic and diverse network visibility in traffic classification. DISTILLER leverages deep learning techniques to automatically extract complex patterns from various modalities of traffic and simultaneously solve multiple traffic categorization problems. The authors evaluate DISTILLER using public datasets and report superior performance compared to state-of-the-art deep learning architectures. Ref. [23] proposes a new multimodal deep learning framework called MIMETIC. MIMETIC overcomes performance limitations by leveraging the diversity of traffic data and achieves superior performance compared to existing single-modal deep learning-based traffic classification methods. It also highlights the effectiveness of multimodal deep learning in classifying traffic by capturing the characteristics of diverse traffic that carry information.

## 3. Proposed Method

In this chapter, we describe three multitasking learning methods that can learn relationships between tasks.

### 3.1. MTC-Based Traffic Classification

3.1.1. Single Task Single Inference

To compare with the proposed three multitask learning methods, we introduce the ST–SI (Single Task–Single Inference) learning method used in existing application traffic classification research. With ST–SI, a main classifier performs the main classification of one task through a single classifier. As shown in Figure 3, ST–SI uses four independent models for the four tasks of browser, protocol, service, and application. Figure 4a shows the structure of the model that performs the browser classification using the extracted features from the flow as the input to the backbone network. The output of the Figure 3a model is one of the three classes of the browser task. Similarly, Figure 3b–d are models responsible for each task, such as protocol, service, and application.

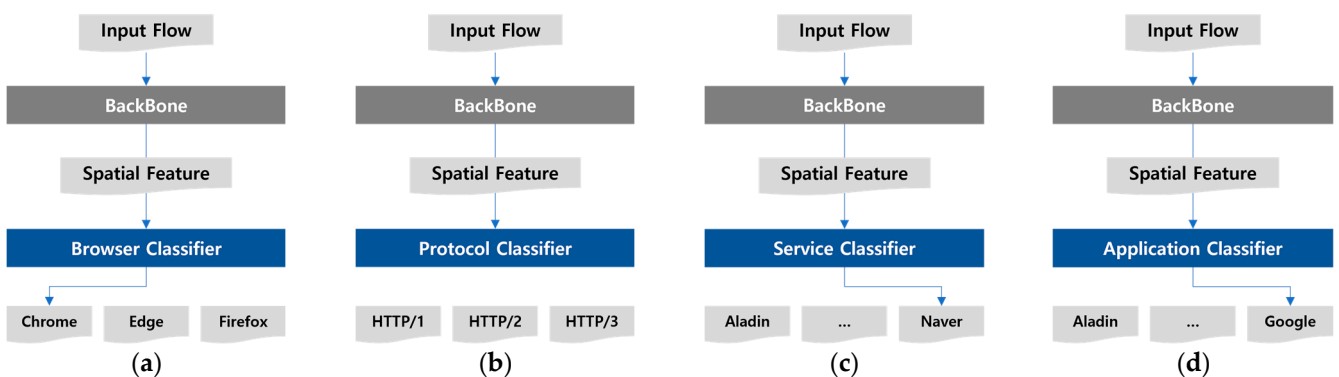

**Figure 3.** Overview of ST–SI (Single Task–Single Inference). (**a**) ST–SI-based browser classification; (**b**) ST–SI-based browser classification; (**c**) ST–SI-based browser classification; and (**d**) ST–SI-based browser classification.

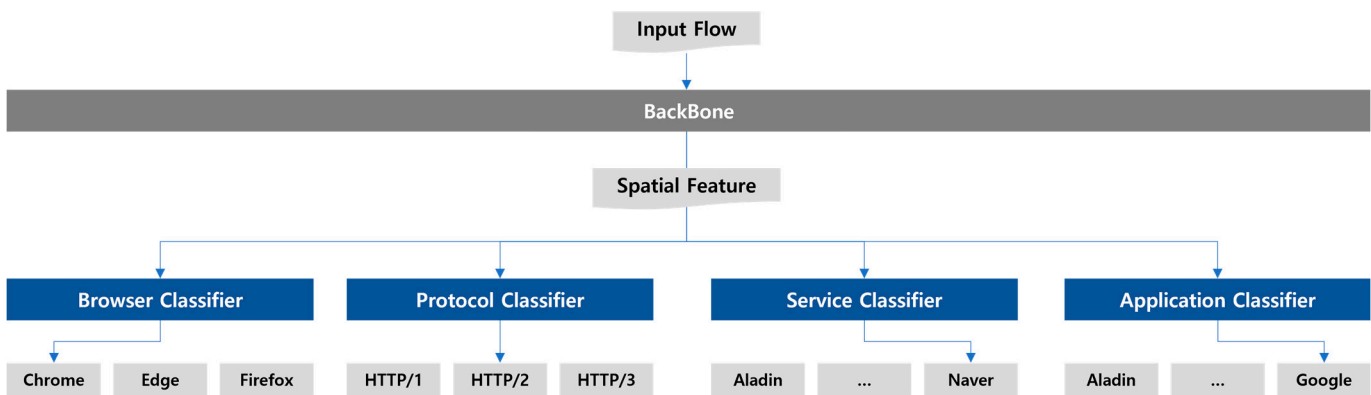

**Figure 4.** Overview of MT (Multitask).

### 3.1.2. Multitask

The first multitask learning method is MT (Multitask), which performs four main classifications simultaneously through four main classifiers. As shown in Figure 4, a single model is used to perform the four tasks of browser, protocol, service, and application. The features extracted from the backbone network are input to each main classifier responsible for the four tasks to predict a single class for each of the four tasks. In this case, each main classifier has an error weight equal to 0.25, meaning that the model learns with equal effort on all four tasks. There may be some common information between the four tasks, and the interactions between them can complement each other and improve performance. Also, by handling multiple tasks and learning common patterns, models that are more robust and flexible for new tasks can be created.

### 3.1.3. Multitask Single Inference

The second multitask learning method is MT–SI (Multitask–Single Inference), which performs the pre-classification of four tasks simultaneously through four pre-classifiers. Then, using the pre-classification results, a main classifier performs the main classification of one task. As shown in Figure 5, the MT–SI learning method uses four independent models for the four tasks of browser, protocol, service, and application. Figure 5a shows the structure of the model that performs the browser classification using the extracted features from the flow as input to the four pre-classifiers and one main classifier. Similarly, Figure 5b–d are models responsible for each task, such as protocol, service, and application. The error weight of the four pre-classifiers in the model is set to 0.1, and the main classifier is set to 0.6. The spatial features generated by the backbone network are input to the pre-classifiers to output classification results (probability of belonging to each class), which are merged with the previously generated spatial features. The main classifier performs the main task using the classification results and and spatial features of four pre-classifiers. The common information shared among tasks extracted during the training process of the pre-classifiers enhances the classification performance of the main classifier.

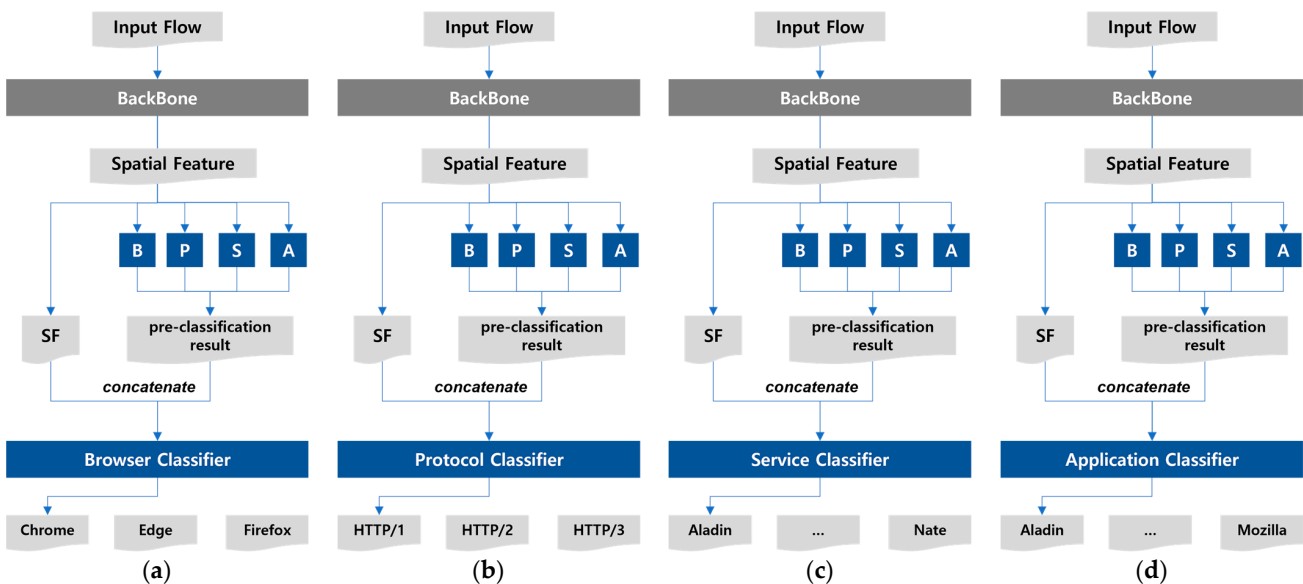

**Figure 5.** Overview of MT–SI (Multitask–Single Inference). (**a**) MT–SI-based browser classification; (**b**) MT–SI-based browser classification; (**c**) MT–SI-based browser classification; and (**d**) MT–SI-based browser classification.

### 3.1.4. Multitask Multi Inference

The third multitask learning method is MT–MI (Multitask–Multi Inference), which performs the pre-classification of four tasks simultaneously through four pre-classifiers. Then, using the pre-classification results, four main classifiers perform the main classification of four tasks simultaneously. As shown in Figure 6, the MT–MI learning method uses a single model for the four tasks of browser, protocol, service, and application. The error weight of each pre-classifier in the model is set to 0.05, and each main classifier is set to 0.2. The spatial features produced by the backbone network are input to the pre-classifiers, which output pre-classification results for each task. The pre-classification results are merged with the previously extracted spatial features and input to each main classifier to perform the corresponding task. The MT–MI method differs from MT in that it performs a brief pre-classification before performing each main classification and uses the results when performing the main classification.

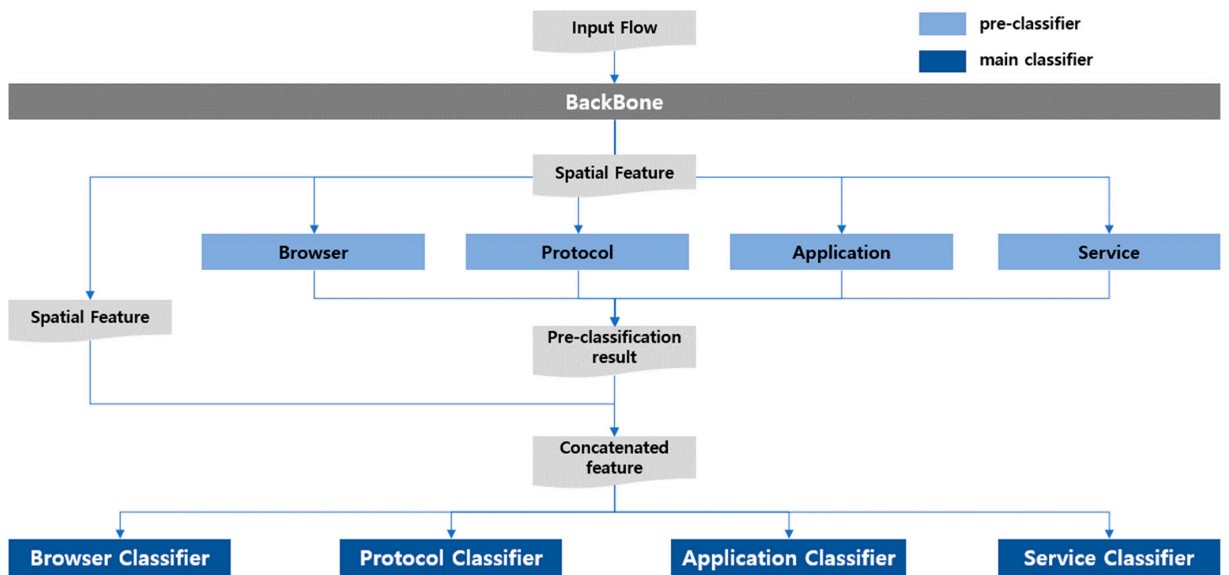

**Figure 6.** Overview of MT–MI (Multitask–Multi Inference).

*3.2. Dataset Description*

This section describes the dataset collected for validating the proposed methodology. When collecting data in a typical environment, traffic unrelated to the target web service can be collected due to the background services that are running. Therefore, a Docker platform that can be isolated from the host network is a good choice. The traffic dataset was collected using Selenium in containers on a Docker platform and consists of six types of web service traffic.

For the three additional tasks, apart from web service traffic, the labeling methods are as follows:

- Service: labeled at the time of collection;
- Browser: labeled at the time of collection;
- HTTP protocol: check the HTTP version of the GET or POST method response header when the protocol of the traffic flow is HTTP (perform the same process after decryption in the case of HTTPS);
- Application: check the Request URL for HTTP or the Service name indicator (SNI) in the Transport Layer Security (TLS) layer for HTTPS.

The collected dataset consists of 10,497 bidirectional flows, and the task-specific distribution is shown in Figure 7. In this figure, the value of each pie is in the form; the number of bi-flows, its percentages. The browser task consists of *Chrome*, *Edge*, and *Firefox*, with *Chrome* and *Firefox* accounting for a high percentage. The protocol task consists of HTTP/1.1, HTTP/2, and HTTP/3, with HTTP/2 accounting for a high percentage. The service task consists of *Aladin*, *Amazon*, *Google*, *Nate*, *Naver*, and *YouTube*, with all six accounting for an equal ratio. The application unit consists of *Aladin*, *Amazon*, *Google*, *Nate*, *Naver*, *YouTube*, *Microsoft*, *Mozilla*, and *Etc*, with *Google* and *Mozilla* accounting for a high percentage. To prevent excessive granularity, labeling was performed for traffic that belongs to major applications, and traffic that does not belong to the eight major applications was assigned to the *Etc* class.

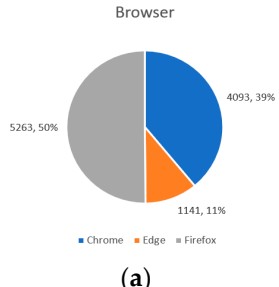
(**a**)

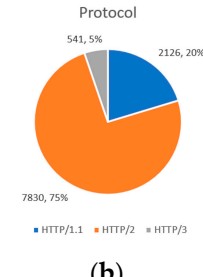
(**b**)

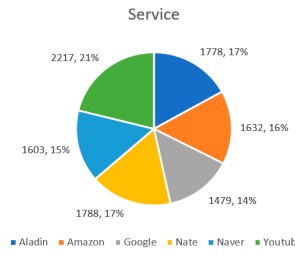
(**c**)

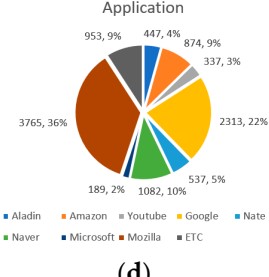
(**d**)

**Figure 7.** The task-specific distribution. (**a**) Browser distribution. (**b**) Protocol distribution. (**c**) Service distribution. (**d**) Application distribution.

Figure 8 shows the protocol ratio by service, with most services primarily using HTTP/2, but *Aladin* and *Nate* services have a relatively high percentage of using the HTTP/1.1 protocol. Also, since the *YouTube* service uses the QUIC protocol, the percentage of using the HTTP/3 protocol is high.

Figure 9 shows the application ratio by browser, and the ratio of applications used by the company that developed each browser is high.

Table 1 shows the application ratio by service, indicating that one or more applications are mixed within a single service traffic.

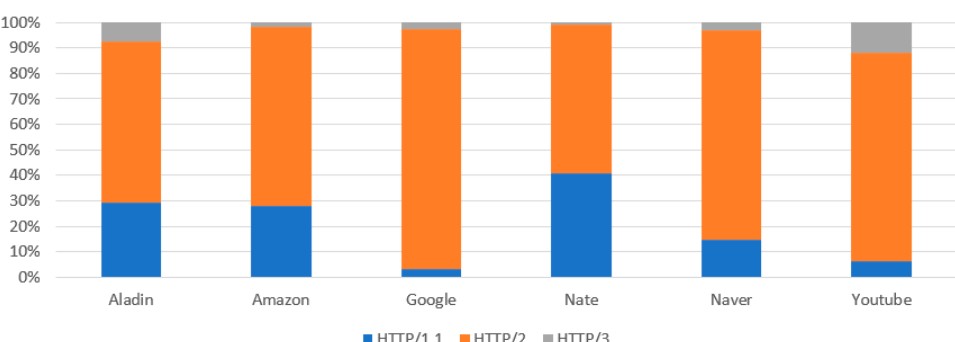

**Figure 8.** The protocol ratio by service.

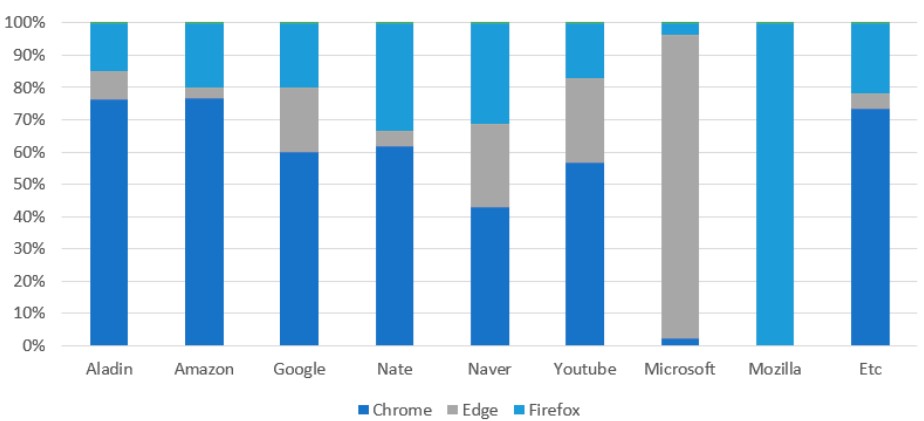

**Figure 9.** The browser ratio by service.

**Table 1.** The application ratio by service.

| Service | | Aladin | Amazon | Google | Nate | Naver | YouTube | Microsoft | Mozilla | Etc |
|---|---|---|---|---|---|---|---|---|---|---|
| Aladin | 1778 100% | 447 25.1% | | 364 20.5% | | 50 2.8% | 5 0.3% | 1 0.1% | 660 37.1% | 251 14.1% |
| Amazon | 1632 100% | | 873 53.5% | 27 1.7% | | | 4 0.2% | 2 0.1% | 612 37.5% | 114 7 |
| Google | 1479 100% | | | 707 47.8% | | | 102 6.9% | 2 0.1% | 668 45.2% | |
| Nate | 1788 100% | | 1 0.1% | 76 4.2% | 537 30% | 10 0.6% | | 2 0.1% | 597 33.4% | 565 31.6% |
| Naver | 1603 100% | | | 17 1.1% | | 1022 63.8% | | 101 6.3% | 451 28.1% | 12 0.7% |
| YouTube | 2217 100% | | | 1122 50.6% | | | 226 10.2% | 81 3.7% | 777 35% | 11 0.5% |

## 4. Experiments

In this chapter, we describe the parameters used in the experiments. The experiments were performed with various parameters, and a total of 576 experiments were conducted by combining five types of parameters.

The first parameter is the four learning methods described in the methodology. The second parameter is the number of packets within a flow, which has three values of 4, 9, and 16. The third parameter is the packet size, which has four values of 324, 400, 484, and 576 (bytes). The fourth parameter is the backbone network, which has four values of LeNet, ResNet, HAST-IDS, and MISCNN. The four backbone networks are used as the backbone network within each learning method during the experiments. HAST-IDS [20] is an intrusion detection system that uses CNN to learn spatial features and LSTM to learn

temporal features. MISCNN is a CNN-based service classification that utilizes various input forms that can be derived from fixed-length packet bytes [19]. The fifth parameter is the input form, which has three values: CP, MP, and MPG. CP (Concatenated Packet input) is an input form that collects and merges the first N bytes of packets within a flow, and extracts features by inputting them to the backbone network, as shown in Figure 10a. MP (Multiple Packet input) is an input form that inputs the first N bytes of packets within a flow to independent backbone networks, and merges the extracted features as shown in Figure 10b. MPG (Multiple Packet input with GRU) has the same form as MP but considers the temporal aspect of packets by inputting the extracted features to GRU, as shown in Figure 10c. A total of 576 experiments are conducted by combining the five parameters.

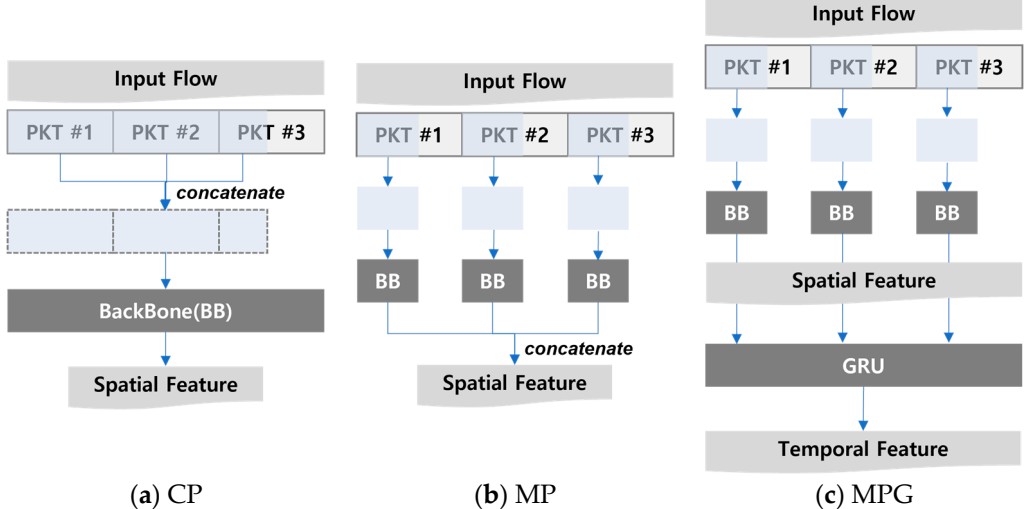

(**a**) CP      (**b**) MP      (**c**) MPG

**Figure 10.** Overview of flow input type. (**a**) Input type CP (Concatenated Packet Input); (**b**) input type MP (Multiple Packet Input); and (**c**) input type MPG (Multiple Packet Input w/GRU).

## 5. Experiment Results

In this chapter, we focus on comparing the classification performance based on the learning method, the number of packets within a flow, the packet size, the backbone network, and the input type.

### 5.1. Comparison of Task Performance According to Parameters

#### 5.1.1. MT Method and Backbone Network

Figure 11 represents the browser classification accuracy based on the multitask approach and backbone network. Figure 11a shows the highest accuracy achieved when applying different experimental parameters to a fixed multitask approach and backbone network. From the perspective of the multitasking approach, methods that utilize the multitasking approach generally achieve higher accuracy compared to the single-task approach, except for when using the Resnet backbone. In terms of the backbone network, Resnet consistently demonstrates good results in browser classification, with the combination of Resnet and MT–SI showing the highest accuracy. Figure 11b represents the standard deviation of the accuracy for combinations of different experimental parameters applied to a fixed multitask approach and backbone network. Overall, experiments that apply the multitask approach tend to have lower standard deviations, indicating that providing prior information to the model is beneficial for generalization. Figure 12 represents the accuracy of HTTP protocol classification based on different multitask approaches and backbone networks. In terms of multitask approaches, except for the case with the Lenet backbone, the method with the application of MT-SI shows higher accuracy. In terms of backbone networks, MISCNN demonstrates overall better results in protocol classification.

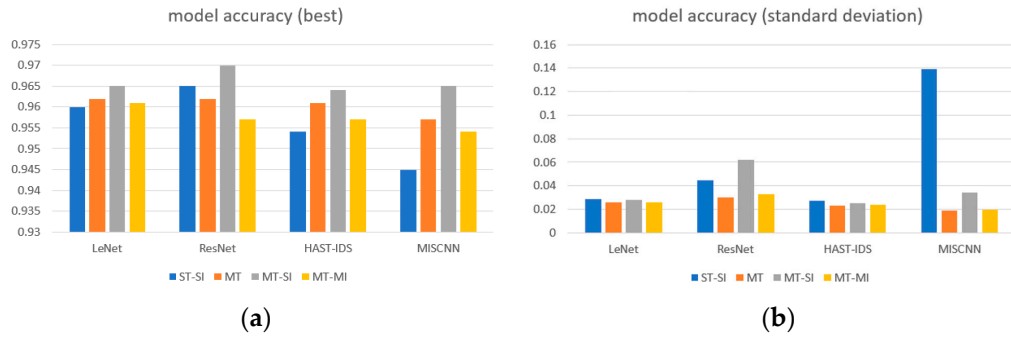

**Figure 11.** Browser classification accuracy according to the MT method and backbone network. (**a**) The highest accuracy among the combinations of experimental parameters; (**b**) standard deviation of the results within the combination. (**a**) Best accuracy. (**b**) The standard deviation of accuracies.

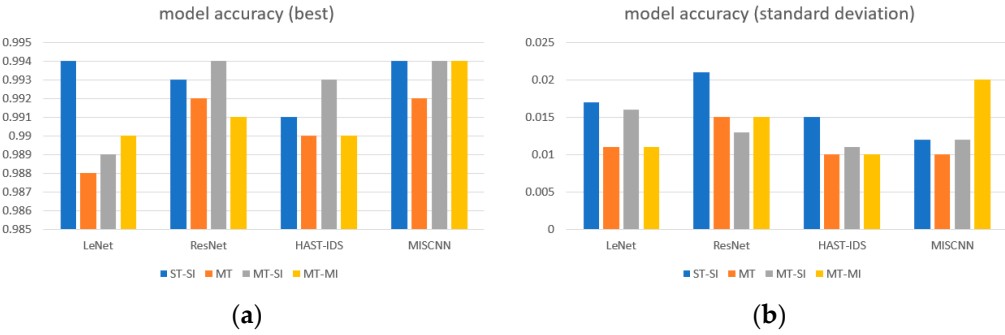

**Figure 12.** Protocol classification accuracy according to the MT method and backbone network. (**a**) Best accuracy. (**b**) The standard deviation of accuracies.

Figure 13 represents the accuracy of service classification based on different multitask approaches and backbone networks. In terms of multitask approaches, except for the case with the Lenet backbone, the method with the application of multitask approaches shows higher accuracy. In terms of backbone networks, HAST-IDS demonstrates overall better results in service classification.

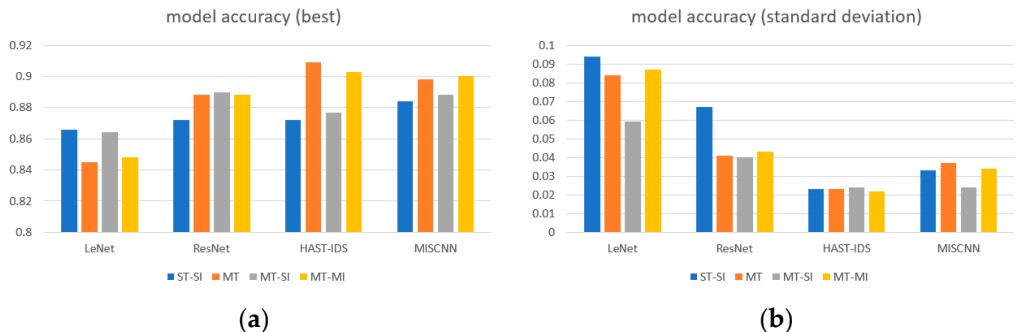

**Figure 13.** Service classification accuracy according to the MT method and backbone network. (**a**) Best accuracy. (**b**) The standard deviation of accuracies.

Figure 14 illustrates the accuracy of application classification based on different multitask approaches and backbone networks. Except for the MISCNN backbone, single-task approaches show better performance. However, it can be observed that when applying multitask approaches, the model's variance is not significant.

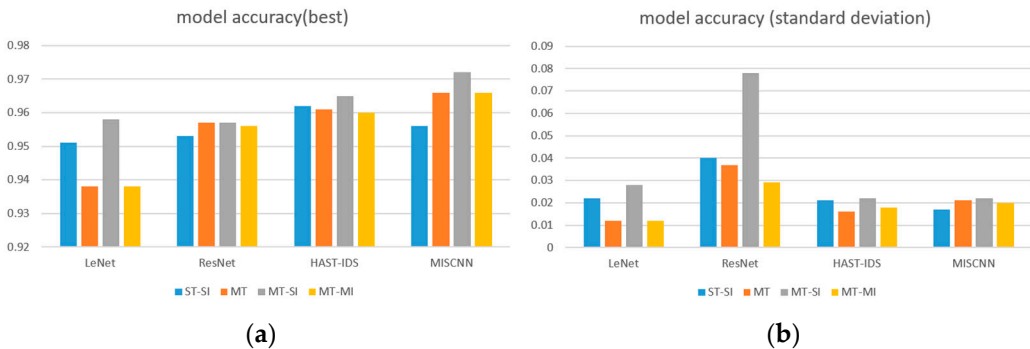

**Figure 14.** Application classification accuracy according to the MT method and backbone network. (**a**) Best accuracy. (**b**) The standard deviation of accuracies.

5.1.2. Performance Comparison by the Number of Tasks

This section describes the performance comparison based on the number of tasks as an additional experiment. Assuming that tasks are not completely independent and have relationships with each other, we compare the performance based on the number of tasks to demonstrate more accurately the classification of other tasks by utilizing the relationships between tasks. We conduct experiments to see if the service classification accuracy improves when the service task is trained with other tasks. To compare the service classification accuracy, the service task is included in all experimental cases. The performance results based on the number of tasks are shown in Table 2. We can observe that the service classification accuracy improves when the service and application tasks are trained simultaneously and when the service, browser, protocol, and application tasks are trained simultaneously. In other words, when training with the application task included, the service classification accuracy improves. Furthermore, we can confirm that the service classification accuracy is the highest when all four tasks are trained simultaneously. Thus, we can see that the service classification task's accuracy improves when trained with other tasks.

**Table 2.** The average accuracy comparison by the number of tasks.

| Task | | | | Accuracy (Service) |
|---|---|---|---|---|
| **Service** | **Browser** | **Protocol** | **Application** | |
| √ | | | | 86.124% ± 0.541% |
| √ | √ | | | 88.698% ± 0.723% |
| √ | | √ | | 89.028% ± 0.613% |
| √ | | | √ | 90.357% ± 0.568% |
| √ | √ | √ | | 89.52% ± 1.195% |
| √ | | √ | √ | 89.81% ± 0.733% |
| √ | √ | | √ | 90.286% ± 0.735% |
| √ | √ | √ | √ | 90.512% ± 0.827% |

*5.2. Ablation Study*

5.2.1. Number of Packets and Backbone Network

Figure 15 illustrates the difference in accuracy based on the change in packet count. In browser and service classification, the accuracy increases as the packet count increases. On the other hand, in protocol and application classification, the accuracy remains similar or decreases as the packet count increases. These results show a consistent trend regardless of the backbone network used.

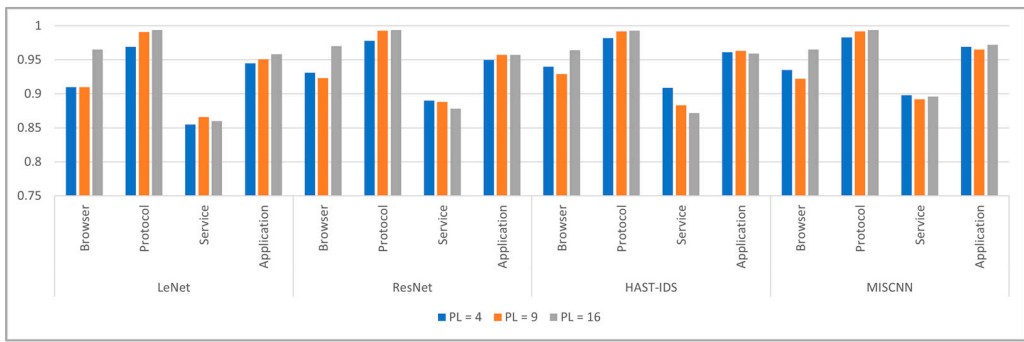

**Figure 15.** Task-specific classification accuracy based on the number of packets and backbone network.

5.2.2. Packet Length and Backbone Network

Figure 16 illustrates the difference in accuracy based on the change in packet length. Except for Lenet, there is not a significant variation in accuracy based on the packet length.

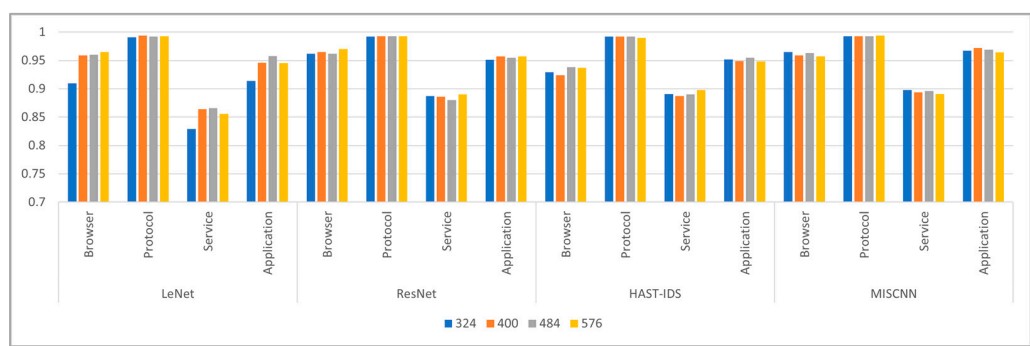

**Figure 16.** Task-specific classification accuracy based on packet length and backbone network.

5.2.3. Input Type and Backbone Network

Figure 17 represents the difference in accuracy based on the input type. In Lenet and Resnet, there is not a significant variation in accuracy based on the input type. However, in HAST-IDS and MISCNN, MPG generally exhibits higher accuracy in most tasks.

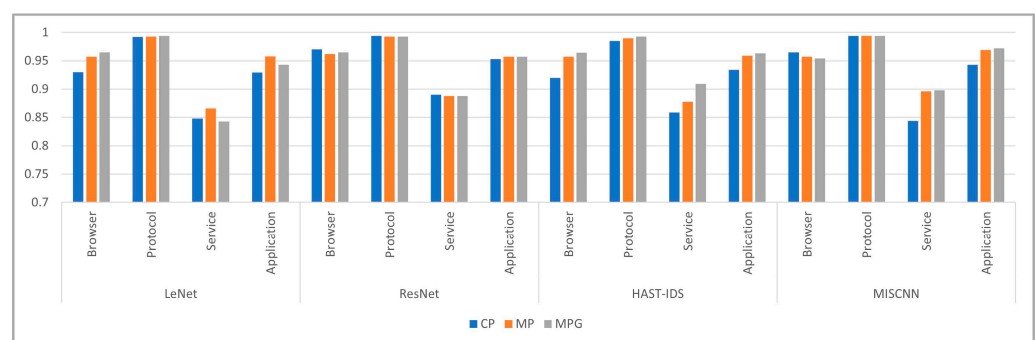

**Figure 17.** Task-specific classification accuracy based on input type and backbone.

5.2.4. Overall

Task-specific overall results based on parameters are shown in Table 3. The following table summarizes the parameters of the model that show the highest performance for each task and backbone network. The multitask learning-based model shows high classification accuracy in the browser, protocol, service, and application tasks, with high accuracy mainly observed between packet lengths of 9 and 16. High accuracy is also observed between packet sizes of 400 and 576, with MP or MPG input type showing high accuracy. When using the multitask learning method for all task classifications, higher classification accuracy is achieved than the conventional ST–SI learning method.

**Table 3.** Best task-specific classification accuracy based on parameters. LN: Lenet; RN: Resnet; HI: HAST-IDS; MC: MISCNN; NP: number of packets; PL: packet length; IT: input type; MT: MT method.

| | Browser | | | | Protocol | | | | Service | | | | Application | | | |
|---|---|---|---|---|---|---|---|---|---|---|---|---|---|---|---|---|
| | LN | RN | HI | MC | LN | RN | HI | MC | LN | RN | HI | MC | LN | RN | HI | MC |
| NP | 16 | 16 | 16 | 16 | 16 | 16 | 16 | 16 | 9 | 4 | 4 | 4 | 16 | 9 or 16 | 9 | 16 |
| PL | 576 | 576 | 484 | 324 | 400 | 400 | 324 | 576 | 484 | 576 | 576 | 324 | 484 | 400 or 576 | 484 | 400 |
| IT | MPG | CP | MPG | CP | MPG | CP | MPG | CP | MP | CP | MPG | MPG | MP | MP or MPG | MPG | MPG |
| MT | MTSI | MTSI | MTSI | MTSI | STSI | MTSI | MTSI | MTSI | STSI | MTSI | MT | MT | MTSI | MT or MTSI | MTSI | MTSI |
| Acc | 0.965 | 0.97 | 0.964 | 0.965 | 0.994 | 0.994 | 0.993 | 0.994 | 0.866 | 0.89 | 0.909 | 0.898 | 0.958 | 0.957 | 0.965 | 0.972 |

### 5.2.5. Confusion Matrix for the Service Task

This section compares the confusion matrices of the service task for ST–SI and MT learning methods. Figure 18a shows the confusion matrix of the service task for the ST–SI learning method. The horizontal axis represents the actual labels, and the vertical axis represents the predicted labels. The result of predicting *YouTube* as *Aladin* in the service task classification using the ST–SI learning method is 8.5, which can be predicted to be a misclassification due to the *YouTube* streaming API call in the *Aladin* product description. Figure 18b shows the confusion matrix of the service task for the MT learning method. The result of predicting *YouTube* as *Aladin* in the service task classification using the MT learning method is 1.2, which has a lower probability of misclassification than the ST–SI learning method. In addition, the result of predicting *YouTube* as *Google* in the service task classification using the ST–SI learning method is 5.7, which can be predicted to be a misclassification due to similar traffic between *Google* and *YouTube* as they belong to the same company's platform. The result of predicting *YouTube* as *Google* in the service task classification using the MT learning method is 0.6, which has a lower probability of misclassification than the ST–SI learning method. The service task confusion matrix shows that the misclassification of *YouTube* into other classes has been improved. Figure 19a shows the confusion matrix of the application task for the ST–SI learning method. The result of predicting *Google* as *YouTube* in the application task classification using the ST–SI learning method is 54.6, which can be predicted to be a misclassification due to similar traffic between *Google* and *YouTube* as they belong to the same company's platform. Figure 19b shows the confusion matrix of the application task for the MT learning method. The result of predicting *Google* as *YouTube* in the application task classification using the MT learning method is 25.8, which has a lower probability of misclassification than the ST–SI learning method. The application task confusion matrix shows that the misclassification of *Google* into *YouTube* has been improved.

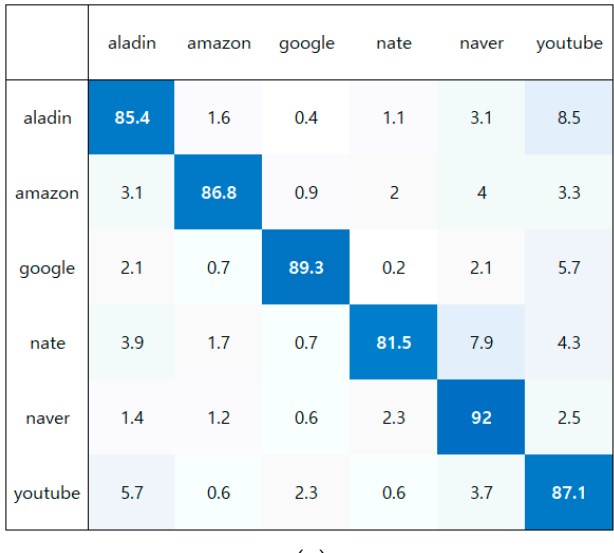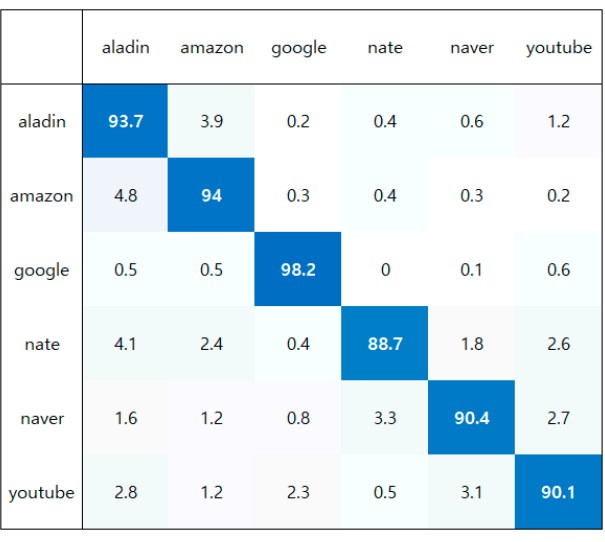

**Figure 18.** Confusion matrix for the service task. (**a**) Confusion matrix for the service task of ST–SI; (**b**) confusion matrix for the service task of MT–SI.

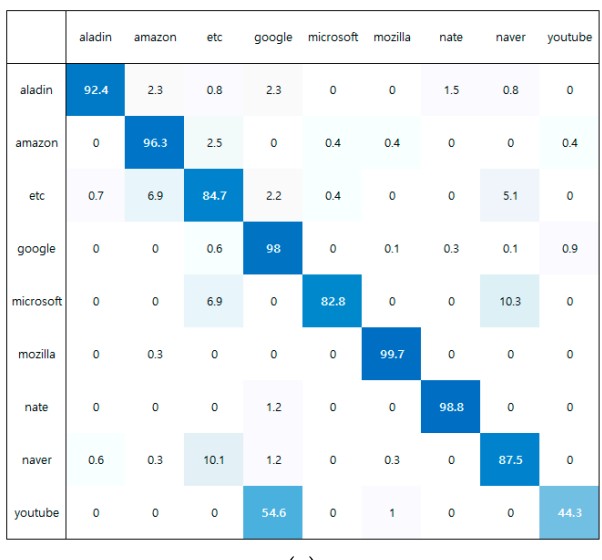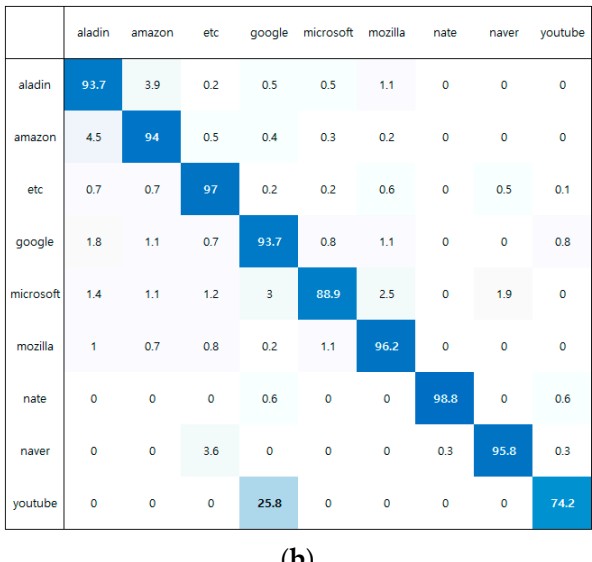

**Figure 19.** Confusion matrix for the application task. (**a**) Confusion matrix for the application task of ST–SI; (**b**) confusion matrix for the application task of MT–SI.

## 6. Conclusions

This paper proposes a multitask learning method for application traffic classification using the relationships between browser, protocol, service, and application tasks. Three multitask learning methods, Multitask, Multitask Single Inference, and Multitask Multi-Inference, are proposed according to the pre-classification status and the number of main classifications. The experimental results show that more accurate application traffic classification can be achieved by utilizing the relationships between tasks. The practicality of the proposed learning methods has been demonstrated through experiments, and accuracy has been improved by 4.4%p on average while maintaining real-time performance. This means that simple model improvements allow network administrators to obtain more accurate classification results without further consideration of the resources of the hardware on which the existing classification model is installed. Specifically, experimental results suggest that if a service is called with another application that does not belong to it, it

is likely to be misclassified in terms of service classification, and that the combination of service and application tasks solves this problem. In addition, applications provided in the form of APIs by *Google* appear to be confusing in classifying other services, but this is also mitigated by the application of multitask learning. Although the combination of service and application tasks has shown high classification accuracy improvements, there is also a small improvement in the combination with the proposed browser and protocol tasks. Also, the multitask learning method can provide accurate and detailed classification results for application traffic, making it widely applicable for various purposes of traffic analysis. Network administrators can receive classification results about the browser (96.5% accuracy), protocol (99.4% accuracy), service (90.9%), and application (97.2% accuracy) of web traffic. Moreover, the experimental results show that the proposed method can be applied to various existing research models. Furthermore, the proposed generalized method of multitasking learning can be combined with state-of-the-art high-performance classification models and has shown high performance in combination with the four backbone networks presented in this study. In future research, we plan to improve the multitask learning method to achieve higher performance by analyzing which classes are difficult to classify in the browser, protocol, service, and application tasks. Several limitations are set for future research. First, the consideration of the gradient conflict problem that may arise in multitask learning has not been addressed. The model's parameters need to be adjusted to satisfy the loss functions for various tasks simultaneously, but the loss functions of different tasks typically have gradients in different directions, which can degrade the performance of multitask learning or make the learning process more difficult. Therefore, there is a need to improve performance through appropriate techniques such as weight sharing or adjustment of loss functions [24]. Second, optimization of the backbone network is another challenge. The backbone network used in this study was only employed to validate multitask learning through newly proposed tasks, so there is a need for model structure adjustments and various parameter adjustments to enhance its capabilities.

**Author Contributions:** Conceptualization, M.-S.K.; methodology, U.-J.B.; software, B.K.; validation, B.K.; formal analysis, U.-J.B.; investigation, J.-T.P. and J.-W.C.; resources, B.K., J.-T.P. and J.-W.C.; data curation, U.-J.B.; writing—original draft preparation, U.-J.B. and B.K.; writing—review and editing, U.-J.B. and M.-S.K.; visualization, J.-W.C.; supervision, M.-S.K.; project administration, M.-S.K.; funding acquisition, M.-S.K. All authors have read and agreed to the published version of the manuscript.

**Funding:** This work was supported by the Technology Innovation Program grant funded By the Ministry of Trade, Industry and Energy (MOTIE, Republic of Korea) and the Korea Evaluation Institute of Industrial Technology (KEIT) (No. 20008902, Development of SaaS SW Management Platform based on 5Channel Discovery technology for IT Cost Saving) and was supported by "Regional Innovation Strategy (RIS)" through the National Research Foundation of Korea(NRF) funded by the Ministry of Education (MOE) (2021RIS-004).

**Data Availability Statement:** The dataset and codes presented in this study are available in request from the primary author https://github.com/pb1069/A-Multi-task-Classification-Method-for-Application-Traffic-Classification-Using-Task-Relationships (accessed on 20 August 2023).

**Conflicts of Interest:** The authors declare no conflict of interest.

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
