# Peer review of "A Multi-Task Classification Method for Application Traffic Classification Using Task Relationships"

_electronics, doi:10.3390/electronics12173597_

Round 1

Reviewer 1 Report

This paper presents a multi-task classification method for application traffic  classification using task relationships, the paper is easy to follow and understand, but there exist some issues that need to be addressed:

1. Both the novelty and technical/scientifical soundness are low:
The main challenges of multi-task learning (MTL) are 1) how to effectively solve the gradient conflict problem during the joint optimization and 2) how to design a backbone to effectively leverage the benefit of MTL. However, I couldn't find sufficient details or new insights regarding the training recipes or model designs.

2. The method section is not very informative: Sec. 3 (Proposed method) is more like a brief introduction to different classification methods, and doesn't provide enough specific details about your proposed method.

3. The experimental results can not effectively validate the superiority of your proposed method:
It would be better to compare your work with the state-of-the-art multi-task classification method;
To ensure the fairness of your comparisons, it would be better to provide your training details. 

4. The quality of figures needs to be improved:
Some figures, particularly in the experiment section, have low resolution, making it challenging to interpret the results.
It is unclear whether Fig. 4 and Fig. 6 are the same or different; clarification on this point would be helpful.

The paper writing is well, but the content is a little bit redundant.

Author Response

Thank you for your detailed review and advice.

Reviewer 2 Report

-       Please, add to the abstract the implications of the findings.

-       Line 33: SNS – the abbreviation is used for the first time. Please, write out the whole word.

-       In the introduction, I suggest relying on the literature to the greater extent. Please, add references to support your arguments.

-       The findings have to be discussed with the findings of the related works to a greater extent.

-       Please, add practical and theoretical implications.

-       I suggest more in-depth conclusions to make based on the findings.

Author Response

(The authors gave the same response as above.)

Reviewer 3 Report

This paper proposes three multi-task learning methods for application traffic classification using the relationships among tasks, composed of browsers, protocols, services, and application units. The proposed methods aim to improve classification performance under the assumption that there are relationships between tasks. The following problems still exist, and the author is suggested to revise them carefully.

1. Please merge the author's unit information, 1,3,4,5, as they are the work units.

2. In the Instruction, what are the main contributions of this paper? It is recommended that the author proposed 2-3 major contributions of this paper.

3. What information does the author want to display through Figure 2 (a)?

4. How is multitasking reflected? Please enhance the description.

5. Figure 14-Figure 19 are too poor and blurry, please improve the quality of the image.

6. The analysis of the research status in this paper is insufficient, some references are too old, and it is recommended to add some references and discussion (about multi-task or classification methods). Such as:

[1] Li, B., Liang, S., Gan, Z., Chen, D., and Gao, P. Research on multi-UAV task decision-making based on improved MADDPG algorithm and transfer learning. International Journal of Bio-Inspired Computation, 2021, 18, 82–91. doi: 10.1504/IJBIC.2021.118087.

[2] Kou, L., Liu, C., Cai, G.-W., Zhou, J.-N., & Yuan, Q.-D. Data-driven design of fault diagnosis for three-phase PWM rectifier using random forests technique with transient synthetic features. IET Power Electronics, 2020, 13(16), 3571–3579. doi: 10.1049/iet-pel.2020.0226

[3] Wilson, B., Dhas, J.P.M., Sreedharan, R.M., and Krish, R.P. Ensemble learning-based classification on local patches from magnetic resonance images to detect iron depositions in the brain. International Journal of Bio-Inspired Computation, 2021, 17, 260–266. doi: 10.1504/IJBIC.2021.116608.

[4] H. Luo, Y. Yang, B. Tong, F. Wu and B. Fan, "Traffic Sign Recognition Using a Multi-Task Convolutional Neural Network," in IEEE Transactions on Intelligent Transportation Systems, vol. 19, no. 4, pp. 1100-1111, April 2018. doi: 10.1109/TITS.2017.2714691

[5] G. Tu, Y. Fu, B. Li, J. Gao, Y. -G. Jiang and X. Xue, "A Multi-Task Neural Approach for Emotion Attribution, Classification, and Summarization," in IEEE Transactions on Multimedia, vol. 22, no. 1, pp. 148-159, Jan. 2020. doi: 10.1109/TMM.2019.2922129

Author Response

(The authors gave the same response as above.)

Round 2

Reviewer 1 Report

Both the novelty and technical/scientifical soundness are limited.

Reviewer 3 Report

Accept